

# DNA Barcoding analysis of seafood accuracy in Washington, D.C. restaurants

David B. Stern[1], Eduardo Castro Nallar[2], Jason Rathod[3] and Keith A. Crandall[1,4]

[1] Computational Biology Institute, Milken Institute School of Public Health, George Washington University, Washington, D.C., USA
[2] Center for Bioinformatics and Integrative Biology, Universidad Andrés Bello, Santiago, Chile
[3] Migliaccio & Rathod LLP, Washington, D.C., USA
[4] Department of Invertebrate Zoology, Smithsonian Institution, Washington, D.C., USA

## ABSTRACT

In Washington D.C., recent legislation authorizes citizens to test if products are properly represented and, if they are not, to bring a lawsuit for the benefit of the general public. Recent studies revealing the widespread phenomenon of seafood substitution across the United States make it a fertile area for consumer protection testing. DNA barcoding provides an accurate and cost-effective way to perform these tests, especially when tissue alone is available making species identification based on morphology impossible. In this study, we sequenced the 5′ barcoding region of the Cytochrome Oxidase I gene for 12 samples of vertebrate and invertebrate food items across six restaurants in Washington, D.C. and used multiple analytical methods to make identifications. These samples included several ambiguous menu listings, sequences with little genetic variation among closely related species and one sequence with no available reference sequence. Despite these challenges, we were able to make identifications for all samples and found that 33% were potentially mislabeled. While we found a high degree of mislabeling, the errors involved closely related species and we did not identify egregious substitutions as have been found in other cities. This study highlights the efficacy of DNA barcoding and robust analyses in identifying seafood items for consumer protection.

## INTRODUCTION

Seafood products are some of the most frequently mislabeled and misrepresented food items across the globe. The DNA-based investigation by Oceana, taking place from 2011 to 2013, revealed that nearly 33% of seafood is mislabeled in retail outlets (*Warner et al., 2013*). Snapper, tuna, and shrimp were the most frequently substituted and often were replaced with species carrying health advisories or of conservation concern (*Warner et al., 2013*). State and federal consumer protection laws prohibit food mislabeling of this kind. One of the strongest such laws is Washington D.C.'s Consumer Protection Procedures Act ("CPPA"). The CPPA is remarkably comprehensive in its protections of consumers and includes expansive remedies if the law is violated. Specifically, the statute allows for individuals to purchase goods for the express purpose of testing their contents to determine whether they comport with the DCCPA. It is illegal under the DCCPA

Corresponding author
Keith A. Crandall, kcrandall@gwu.edu

to "represent that goods or services have a source, sponsorship, approval, certification, accessories, characteristics, ingredients, uses, benefits, or quantities that they do not have" (D.C. Code § 28-3904). Violations of the DCCPA entitle plaintiffs to $1500 in statutory damages per violation and the ability to recover not just for themselves, but on behalf of the general public, enabling citizens to act as "private attorneys general" (*Rubenstein, 2004*).

DNA barcoding is a particularly useful tool for making species-level identifications when other data (morphological, geographical, ecological, etc.) are not available or may mislead identifications. The 5′ end of the Cytochrome Oxidase I (COI) mitochondrial gene has been established as the "barcode" sequence for metazoan taxa due to its high variability and conservation of PCR primer sites (*Hebert et al., 2003*). Due to the efforts of biologists around the world, there is a large database of barcode DNA sequences from 260,654 species of animals, plants, fungi and other life (Barcode of Life Database, http://www.boldsystems.org/, Accessed December 1, 2016), facilitating identification. The approach has been proven to be a valuable forensic tool in identifying substitution in the seafood market (*Ardura et al., 2010*; *Hanner et al., 2011*; *Lowenstein, Amato & Kolokotronis, 2009*; *Torres et al., 2013*; *Vandamme et al., 2016*; *Wong & Hanner, 2008*), as well as identifying seafood that carry conservation and human health concerns (*Lowenstein et al., 2010*). Although sophisticated uses of DNA barcoding exist to characterize large dataset's of DNA barcodes, (e.g., *Brown et al., 2012*; *Zhang et al., 2012*), a typical barcoding analysis consists of querying a sampled sequence against a database such as the National Center for Biotechnology Information (NCBI) GenBank or BOLD using BLAST (*Altschul et al., 1990*; *Altschul et al., 1997*) or a similar algorithm. The potential pitfalls of using this method to identify species have been explored and include sequencing errors, lack of variation in the barcode marker, incorrect identification of voucher sequences, arbitrary similarity cutoffs, unavailability of references and accidental sequencing of nuclear mitochondrial genes, in addition to methodological and theoretical shortcomings (*Collins & Cruickshank, 2013*).

Even beyond seafood and species identification, phylogenetic methods have a history of use in US court cases (e.g., *Metzker et al., 2002*). For our purposes, phylogenetics presents a potential to increase and supplement the strength of similarity-based identifications by setting the tests in a statistical framework (*Huelsenbeck & Crandall, 1997*). When 100% identical matches are not found or when multiple reference sequences are similarly distant from the query, phylogenetic topology tests can garner statistical support to reject alternative hypotheses. We can test *a priori* hypotheses based solely on the labeling of the seafood item, or test if an identification made by some other method is significantly better than a competing one.

In this study, as part of a preliminary legal investigation exploring the viability of a lawsuit under Washington D.C.'s CPPA, we sample the barcoding region of the COI gene to test the identity of commonly mislabeled seafood products from six restaurants in Washington, D.C. We take multiple approaches to identify the sampled specimens. While we report multiple potentially mislabeled food items, the majority of our tested items were either correctly labeled or were identified as legally acceptable species for the food labeling.

## MATERIALS AND METHODS

### Tissue and data collection

Six restaurants in Washington, D.C. (Table 1) were visited in March of 2015. Two seafood dishes were sampled from each restaurant, targeting menu-listings with potential human health or conservation concerns. One tissue sample per menu item was stored in 95% ethanol. Care was taken to only collect tissue from the main dish from the center of the food item after rinsing away other substances. Total DNA from each sample was extracted with the Qiagen DNeasy DNA extraction kit. PCR amplification of COI was performed using the LCO1490/HCO2198 (*Folmer et al., 1994*) primers for invertebrate species and FishF2_t1/FishR2_t1 (*Ivanova et al., 2007*) for vertebrate species. 25 μl PCR reactions contained 2.5 μl 10X PCR Buffer, 0.75 μl 50 mM MgCl$_2$, 0.5 μl 10 mM dNTPs, 0.5 μl 10 μM forward primer, 0.5 μl 10 μM reverse primer, 0.1 μl Platinum *Taq* DNA Polymerase (Thermo Fisher Scientific, Waltham, MA, USA) and 1 μl DNA template. After initial denaturation step of 95 °C for five minutes, PCR took place in 35 cycles of 95° for 30 s, 50° for 45 s and 72° for 60 s with a final extension of 72° for five minutes. PCR products were cleaned with ExoSAP-IT (Affymetrix, Santa Clara, CA). Each product was sequenced in both directions using the Big Dye (Life Technologies) cycle sequencing protocol and an ABI 3730XL sequencer. We followed *Song et al. (2008)* to avoid sequencing nuclear copies of COI by checking for high-quality base calls and assuring the absence indels and stop codons in sequence alignments. Sequences have been deposited to NCBI GenBank for public access (Accession numbers: KY656473–KY656484).

### Database searches and reference sequence collection

Sequences were searched against the Barcode of Life Database (http://www.boldsystems.org) for initial species identification. BOLD contains barcode COI sequences from NCBI as well as from other researchers and institutions who deposit sequences along with species identifications and metadata. We also collected COI sequences from all species present in the top 100 hits of the NCBI BLAST searches and BOLD searches, if available. Multiple reference COI sequences for the reported species or taxonomic group based on the menu listings were collected from NCBI GenBank. Two types of situations arose in which a proper reference sequence was not available. One was where the menu listing makes reference to a particular species, but a COI sequence was not available on NCBI or BOLD (e.g., Rock Shrimp, *Sicyonia brevirostris*). In this situation, we collected sequences from related species based on NCBI taxonomy. The other was where the menu listing does not reference one particular species (e.g., Snapper). In this situation, we used only the top BLAST hits from NCBI and BOLD in subsequent analyses.

### Phylogenetic analysis

Two alignments were generated, one each for invertebrate and vertebrate sequences. Sequences were aligned using TranslatorX (*Abascal, Zardoya & Telford, 2010*) which translates protein-coding sequences to amino acids, aligns protein sequences with MAFFT v7.305b (*Katoh & Standley, 2013*) and then back-translates to nucleotides. This aids in the

**Table 1** Sampling information and identification with BOLD and the maximum-likelihood phylogeny.

| Restaurant | Menu Listing | Putative Genus | Putative Species | BOLD ID (Top percent identity) | ML COI Phylogeny ID (Bootstrap support/ PP of node) |
|---|---|---|---|---|---|
| Bobby Van's Steakhouse | Rock Shrimp Tempura | *Sicyonia* | *brevirostris* | *Litopenaeus vannamei* (100) | *Litopenaeus vannamei* (88/87) |
| | Calamari | NA | NA | *Uroteuthis edulis* (100) | *Uroteuthis edulis* (98/100) |
| Gordon Biersch | Yellowfin Tuna | *Thunnus* | *albacares* | *Thunnus albacares* (100), *Thunnus atlanticus* (100), *Thunnus obesus* (100) | *Thunnus albacares* (11/6) |
| | Gulf Shrimp | NA | NA | *Farfantepenaeus aztecus* (99.85) | *Farfantepenaeus aztecus* (100/100) |
| The Oceanaire Seafood Room | Chilean Seabass | *Dissostichus* | *eleginoides* | *Dissostichus mawsoni* (100) | *Dissostichus mawsoni* (99/100) |
| | Australian Barramundi | *Lates* | *calcarifer* | *Lates uwisara* (100), *Lates calcarifer* (100) | *Lates calcarifer* (100/99) |
| Joe's Seafood, Prime Steak and Snow Crab | Ahi Tuna Tartare | *Thunnus* | *albacares* | *Thunnus alalunga* (100), *Thunnus obesus* (99.83), *Thunnus orientalis* (99.67), *Thunnus maccoyii* (99.67) | *Thunnus* sp. (31/9) |
| | Chilean Seabass | *Dissostichus* | *eleginoides* | *Dissostichus eleginoides* (100) | *Dissostichus eleginoides* (99/99) |
| Legal Sea Foods | Snapper Salsa Verde | *Lutjanus* | NA | *Lutjanus guttatus* (100), *Lutjanus* sp. (100), *Lutjanus synagris* (98.31) | *Lutjanus guttatus, Lutjanus synagris* (85/41) |
| | Everything Tuna | *Thunnus* | NA | *Thunnus obesus* (100), *Thunnus albacares* (99.84) | *Thunnus obesus* (61/52) |
| McCormick and Schmick's | Sesame Crusted Albacore Tuna | *Thunnus* | *alalunga* | *Thunnus albacares* (100), *Thunnus atlanticus* (100), *Thunnus obesus* (100) | *Thunnus albacares* (14/6) |
| | Pesto Chilean Seabass | *Dissostichus* | *eleginoides* | *Dissostichus eleginoides* (100) | *Dissostichus eleginoides* (99/99) |

identification and removal of sequences with premature stop codons and indels that are indicative of numts (*Song et al., 2008*).

Best-fit models of evolution and an optimal data-partitioning scheme were chosen using PartitionFinder v1.1.1 (*Lanfear et al., 2012*) with each codon position chosen as *a priori* data subsets and using the Bayesian Information Criterion (BIC) for model selection (*Posada & Crandall, 2001*). A maximum-likelihood tree was estimated in RAxML 8.2 (*Stamatakis, 2014*) using the partitioning scheme selected with PartitionFinder. 1,000 bootstrap replicates were followed by 10 maximum likelihood tree searches under the GTRCAT model with final optimization under GTRGAMMA. We used MrBayes 3.2.2 (*Ronquist et al., 2012*) to asses the posterior probabilities of nodes recovered in the maximum likelihood estimate using the partitioning scheme and models suggested by PartitionFinder. Four independent runs of four MCMC chains were run for 10,000,000 generations, sampling the cold chain every 1000 steps for a total of 40,000 samples. The first 10% of samples were discarded as burn-in. Convergence was assessed by assuring the standard deviation of split frequencies across runs was below 0.01 and that ESS values for all parameters were above 10,000.

## Topology tests

A strength of phylogenetic analysis comes from the ability to compare statistical support for alternative topologies (*Huelsenbeck & Crandall, 1997*). In this way, we can test alternative *a priori* hypotheses, where appropriate, in which the sampled sequences are constrained to form monophyletic groups with reference sequences or constrained to not form monophyletic groups with those sequences. While monophyly does not equate to identification, strong support against monophyly would suggest that the query sequence is not the same species as the target group. We formed the alternative hypotheses based on the menu listings (Table 2). For example, we were able to test if our Chilean Seabass sequence formed a monophyletic group with other Chilean Seabass sequences, but could only test if "Snapper" formed a monophyletic group with other sequences that could be labeled "Snapper," since the menu listing was not species-specific. Where appropriate, we also tested hypotheses for alternative groupings based on the maximum likelihood phylogeny. In a Bayesian framework, model selection is performed using the Bayes factor, which is the ratio of marginal likelihoods of two competing hypotheses. Marginal likelihoods were estimated for each hypothesis using stepping-stone analysis in MrBayes 3.2.2 (*Ronquist et al., 2012*) with the same models and partitioning scheme as above. Stepping-stone analyses were executed with four independent runs of four chains each for 50 steps of 200,000 generations each (sampled every 1,000 generations) each for a total of 10,000,000 generations in each run for each hypothesis. The first step (200,000 generations) was discarded as burn-in. We considered a Bayes factor greater than 5 as strong support for one hypothesis over another (*Kass & Raftery, 1995*).

## Character-based tests

Fish species of the genus *Thunnus* have been shown to have considerably low levels of COI diversity (*Ward et al., 2005*) potentially as the result of rapid speciation and large effective

**Table 2  Tested hypotheses and marginal log likelihoods used for Bayes Factors.**

| Restaurant | Menu listing | Tested hypotheses (Marginal LOG likelihood) |
|---|---|---|
| Bobby Van's Steakhouse | Rock Shrimp Tempura | Monophyletic with *Sicyonia* (−6984.27), Not monophyletic with *Sicyonia* (−6951.31), **Monophyletic with Whiteleg Shrimp (−6940.03)** |
| Gordon Biersch | Yellowfin Tuna | **Monophyletic with Yellowfin Tuna (−5463.35),** Not-Monophyletic with Yellowfin Tuna (−5471.71) |
| The Oceanaire | Chilean Seabass | Monophyletic with Chilean Seabass (−5532.76), Not monophyletic with Chilean Seabass (−5469.00), **Monophyletic with Antarctic Toothfish (−5457.31)** |
| | Australian Barramundi | **Monophyletic with Australian Barramundi(−5458.50),** Not monophyletic with Australian Barramundi −5472.81) |
| Joe's Seafood, Prime Steak and Snow Crab | Ahi Tuna Tartare | Monophyletic with Ahi (−5485.01), Not monophyletic with Ahi (−5472.92), **Monophyletic with Albacore (−5460.75)** |
| | Chilean Seabass | **Monophyletic with Chilean Seabass(−5460.67),** Not monophyletic with Chilean Seabass(−5470.68), Monophyletic with Antarctic Toothfish (−5530.80) |
| Legal Sea Foods | Snapper Salsa Verde | **Monophyletic with *Lutjanus* (−5432.61),** Not monophyletic with *Lutjanus* (−5490.42), Monophyletic with *L. synagris* (−5460.98), Monophyletic with *L. guttatis* (−5460.73) |
| McCormick and Schmick's | Sesame Crusted Albacore Tuna | Monophyletic with Albacore (−5492.50), Not monophyletic with Albacore (−5471.55), **Monophyletic with Yellowfin (−5461.95)** |
| | Pesto Chilean Seabass | **Monophyletic with Chilean Seabass(−5460.35),** Not monophyletic with Chilean Seabass(−5531.72), Monophyletic with Antarctic Toothfish (−5469.87) |

population sizes that make distance-based identifications challenging (*Elias et al., 2007*). In this situation, a query sequence can have a nearly identical distance to multiple different reference sequences. While model-based phylogenetic approaches can be effective in this case, we also took advantage of the character-based identification key of *Lowenstein, Amato & Kolokotronis (2009)* to identify tuna samples. This key consists of 14 nucleotide characters at specific positions across the COI barcoding region developed from an alignment of 87 reference sequences with diagnostic states that are specific to each of the eight tuna species.

## RESULTS

### Database searches

BOLD searches were able to make species-level identifications for six of the 12 sampled specimens and genus-level identifications for the other six (Table 1). BOLD did not make species-level identifications for "Australian Barramundi," "Snapper Salsa Verde" or any of the tuna samples. The "Australian Barramundi" sample was a 100% identical match with *Lates uwisara* and *L. calcarifer*. The BOLD records for *Lates uwisara* linked to GenBank records that were named as *L. calcarifer* and were therefore treated as *L. calcarifer* in phylogenetic analyses. Of the six species-level identifications, two did not match the species reported by the restaurants (Table 1). The COI sequence generated from

Bobby Van's "Rock Shrimp Tempura" sample matched the Whiteleg shrimp (*Litopenaeus vannamei*) sequence with 100% identity (Table 1). The Chilean Seabass sequence from The Oceanaire Seafood Room matched the Antarctic Toothfish (*Dissostichus mawsoni*) with 100% similarity.

While the labeling of "Gulf shrimp," "Calamari," "Everything Tuna" and "Snapper Salsa Verde" are not species-specific, the "Gulf shrimp" from Gordon Biersch matched *Farfantepenaeus aztecus* with up to 99.85% similarity and the "Calamari" from Bobby Van's Steakhouse matched *Uroteuthis edulis* with 100% identity. The "Everything Tuna" sample matched three different tuna species with *Thunnus obesus* (Bigeye Tuna) as the best hit and "Snapper Salsa Verde" matched three different species in the genus *Lutjanus* with *L. guttatus* as the best hit.

## Phylogeny-based tests

The final alignments for vertebrate and invertebrate taxa were 699 and 708 bp long, respectively, and contained no gaps. Inspecting placement of query sequences in the maximum likelihood COI trees supported the BOLD identifications with high bootstrap support and posterior probability, except in the case of "tuna" samples, where the phylogeny failed to resolve relationships with high support and numerous species were not monophyletic (Figs. 1 & 2). The failure to resolve a number of species and genera as monophyletic was likely the result of relying on a single mitochondrial marker to estimate a phylogeny of many distantly related groups. This is expected given the numerous biological and methodological reasons that a given gene tree may not match the true "species" tree (*Edwards, 2009*).

Bayesian topology tests were able to garner support for identifications where BOLD searches, based on percent similarity, could not (Table 2). Our analysis of Joe's Seafood's "Ahi Tuna" suggested with strong support (BF > 15) that the sequence was not monophyletic with other Ahi (Yellowfin, *Thunnus albacares*) sequences. The maximum-likelihood phylogeny placed this sequence sister to a clade of six other tuna species and our Bayesian topology test supported its grouping with Albacore tuna (*Thunnus alalunga*) with strong support over its grouping with Yellowfin/Ahi (BF∼25). BOLD analysis suggested that the Yellowfin tuna from Gordon Biersch could have been one of three species (Table 1) and the maximum-likelihood phylogeny narrowed this down to a likely grouping with Yellowfin (*T. albacares*). The topology test agreed (BF = 8.36) that the sequence formed a monophyletic group with other Yellowfin sequences. We found strong support for the hypothesis that our sample from McCormick and Schmick's "Albacore Tuna" did not form a monophyletic group with the other Albacore (*Thunnus alalunga*) sequences (BF = 20.95). The maximum-likelihood phylogeny placed this sequence sister to a *Thunnus albacares* sequence and our topology test supported this grouping over *Thunnus alalunga* (BF = 24.55). The maximum-likelihood phylogeny grouped the "Snapper Salsa Verde" sequence with an assemblage of *Lutjanus guttatus* and *L. synagris*. While the topology tests suggested strongly that this sequence grouped with other *Lutjanus* sequences (BF > 55),
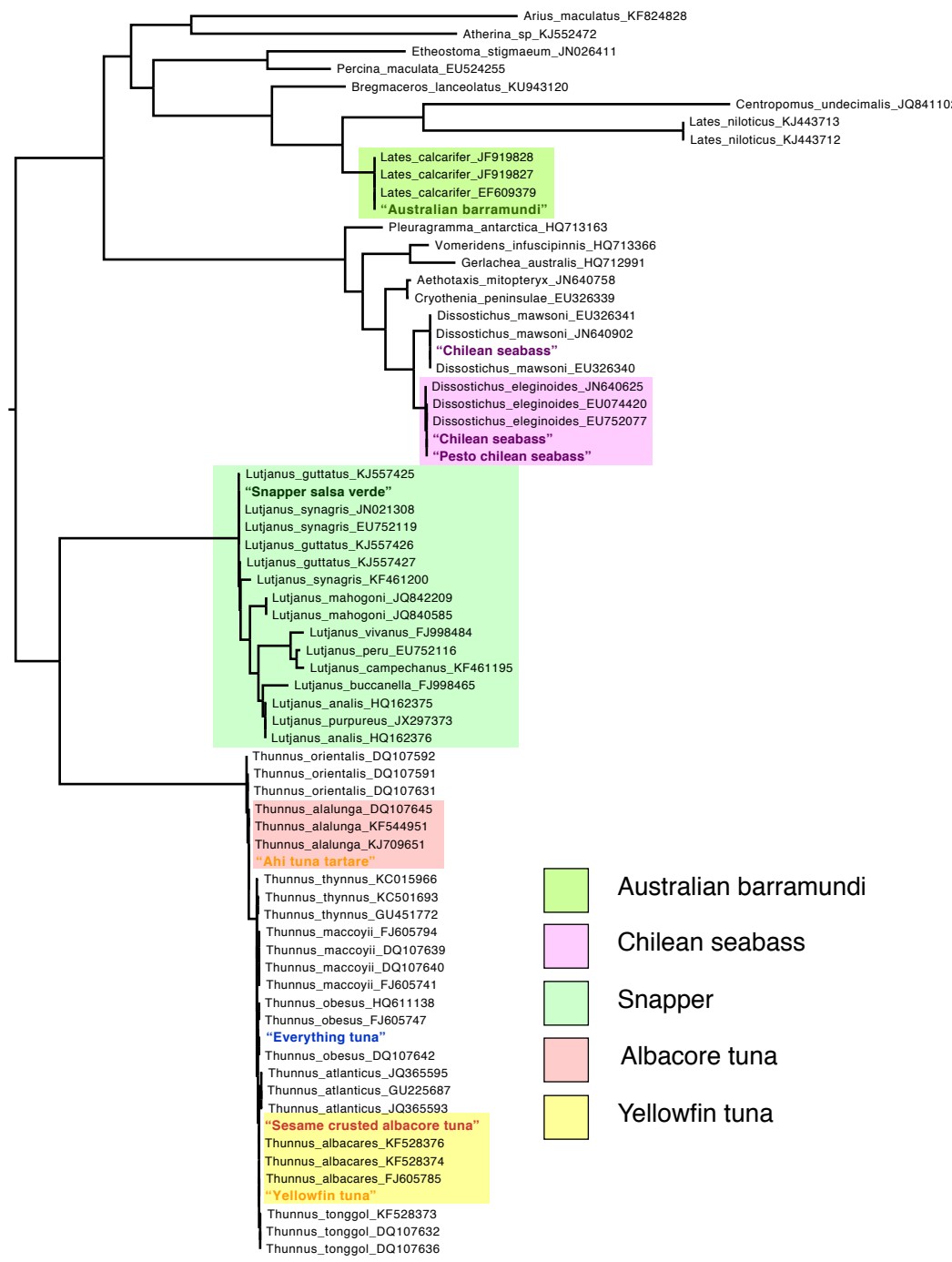

**Figure 1  Maximum-likelihood phylogeny of Vertebrate COI sequences.** Names of samples from this study are colored according to the "target" sequences of the menu-listing. Reference sequence labels include GenBank accession numbers. Sequences that are only found on BOLD are labeled as such before the accession number.

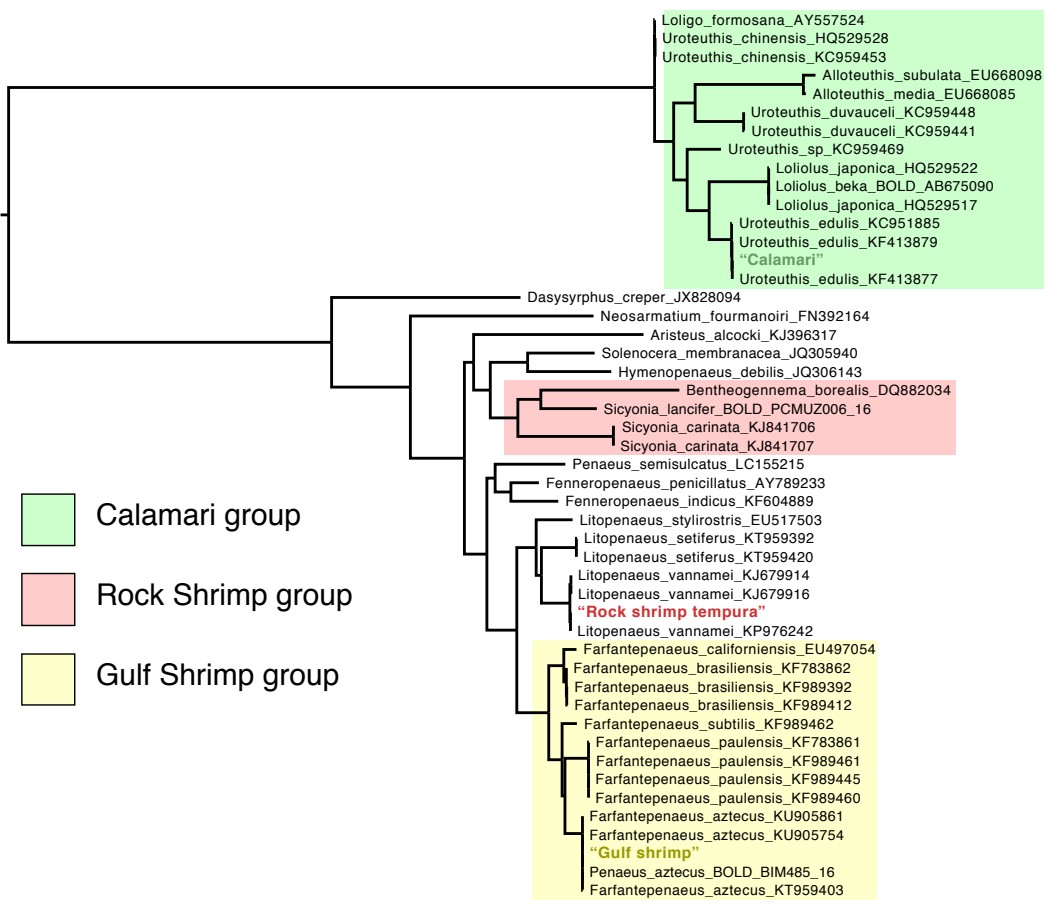

**Figure 2** Maximum-likelihood phylogeny of Invertebrate COI sequences.

we could not find significant support for its grouping with *Lutjanus guttatus* or *L. synagris* over the other (BF < 1).

Regardless of the fact that there was no reference COI sequence for Rock Shrimp (*Sicyonia brevirostris*) available for download, we rejected the hypothesis that this sequence formed a group with other *Sicyonia* sequences and found strong support for its grouping with the Whiteleg Shrimp (*Litopenaeus vannamei*, BF > 40).

### Character-based identification

Using the character-based identification scheme of *Lowenstein, Amato & Kolokotronis (2009)*, we were able to make species-level identifications of all of the tuna specimens (Table 3). Only one of the four tuna specimens matched the menu-listed species, although one did not make a species-level listing. The character-based identifications matched our phylogenetic hypotheses in all cases.

## DISCUSSION

DNA barcoding is a powerful and cost-effective method for making species-level identifications of seafood items. Here we take advantage of multiple analytical methods

Stern et al. (2017), *PeerJ*, DOI 10.7717/peerj.3234

**Table 3  Character based identification for tuna species.**

| Restaurant | Menu listing | Character positions (*Lowenstein, Amato & Kolokotronis, 2009*) (262, 268, 271, 286, 313, 337, 358, 400, 409, 475, 478, 484, 508, 535) | ID |
|---|---|---|---|
| Gordon Biersch | Yellowfin tuna | CCCC ACGT ATTG AC | Yellowfin tuna (*T. albacares*) |
| Joe's Seafood, Prime Steak and Snow Crab | Ahi Tuna Tartare | CTCC GCAT ATCA AT | Albacore (*T. alalunga*) |
| Legal Sea Foods | Everything Tuna | CCCT ACGG ATTG AC | Bigeye tuna (*T. obsesus*) |
| McCormick and Schmick's | Sesame Crusted Albacore Tuna | CCCC ACGT ATTG AC | Yellowfin tuna (*T albacares*) |

(database searches, phylogeny estimate, topology tests, character-based ID) using DNA barcoding to test for seafood substitutions in six restaurants in Washington, D.C. Perhaps the clearest act of seafood substitution was the sale of Whiteleg shrimp (*Litopenaeus vannemei*) as "Rock Shrimp." This was a commonly observed trend in Oceana's 2014 study (*Warner et al., 2014*). One of the three "Chilean seabass" we tested was identified as Antarctic toothfish, the sister species to the "Chilean seabass" also known as the "Patagonian toothfish." In general there was agreement, where possible, across multiple methods, and where BOLD was unable to make a species-level identification, another method was able to narrow the identification further. Three of the major issues confronted in this analysis were (1) lack of available reference sequence, (2) ambiguous menu listings, and (3) lack of sequence variation among congeneric species.

The lack of available reference sequences was largely an issue that was discovered through database searches and phylogeny estimation. Particularly for the "Rock Shrimp" specimen, there are no available reference COI sequences for *Sicyonia brevirostris* on GenBank or BOLD. Therefore, we were not able to explicitly test if our sampled "Rock Shrimp" sequence was *Sicyonia brevirostris* even though BOLD does have sequences from several *Sicyonia* species including *S. brevirostris*. It is unfortunate that these are not currently made available to the public especially considering the widespread sale and consumption of "Rock Shrimp" across the country (*Warner et al., 2014*). We were able to test if our sample belonged to the genus *Sicyonia* because COI sequences from *S. carinata* and *S. lancifer* were available. Nevertheless, the sequence matched with 100% identity to the Whiteleg Shimp, *Litopenaeus vannamei.*

Many of the menu listings were vague enough to allow an array of acceptable species. For example, "snapper" can refer to any of the 113 species in the family Lutjanidae or to species from a number of other families with a common name that includes "snapper." "Red snapper" is approved in the US to only refer to the species *Lutjanus campechanus* and has been involved in a number of cases of seafood fraud (*Marko et al., 2004*). The phylogenetic analyses strongly supported our sample's placement in the genus *Lutjanus,* but we were not able to distinguish between an identification of *L. guttatus* or *L. synagris.* Either these species are very closely related (resulting in little variation among COI sequences) or reference sequences were not properly identified. Our analyses strongly suggested that the "Calamari" specimen was *Uroteuthis edulis,* which is one of many squid species approved to be sold as "Calamari" by the FDA (http://www.accessdata.fda.gov, Accessed August 25, 2016). *Farfantepenaeus* shrimp species are commercially important with millions of pounds being caught and sold each year (*Warner et al., 2014*). *Farfantepenaeus aztecus*, also known as "brown shrimp," is the most widely fished species in the genus. These species are found in the Gulf of Mexico and are therefore often referred to as "Gulf Shrimp."

There are eight recognized species in the genus *Thunnus*, but all can be sold under the name "tuna" or its synonyms in the US according to the FDA (*Lowenstein, Amato & Kolokotronis, 2009*). Database searches and phylogeny estimation failed to make species-level identifications for our "tuna" samples based on the COI sequence. The very low variability among tuna COI sequences has been observed before and results in low support for phylogenetic groupings (*Ward et al., 2005*). Phylogenetic topology tests did prove

more successful in this manner, because we were able to directly compare support for alternative hypotheses, even if the probability of one particular phylogeny or grouping is not objectively high. These results agreed with the character-based identifications, which proved to be the most useful for tuna samples in combination with phylogenetic tests.

The "Everything Tuna" sample was identified as *Thunnus obesus*, which is given a status of "Vulnerable" by the International Union for Conservation of Nature Redlist (*IUCN, 2015*). This was the only one of our samples that was identified as a species with a potential conservation concern. This was certainly lower than the rate of threatened or endangered species found in other cities that have been surveyed, although we did not test grouper, halibut or eel samples, which are frequently substituted with fish from threatened populations (*Vandamme et al., 2016*; *Warner et al., 2013*). While none of the substituted seafood items we found carry well-known health risks or conservation concerns *per se*, the labeling of farmed Whiteleg Shrimp as "Rock Shrimp" is worrisome as wild-caught shrimp are not required to be screened for veterinary drug residue levels like farmed shrimp (*US Food and Drug Administration, 2001*).

## CONCLUSIONS

Of our 12 samples from six restaurants, we found that four menu items, one "Chilean Seabass," two "Tuna" and one "Rock Shrimp," were potentially mislabeled, albeit with species that are either closely related or typically considered acceptable for the menu listing. This is consistent with the 33% average rate found across United States cities by Oceana and lower than seven of the 12 cities surveyed in that study (although higher than the 26% found in Washington, D.C. by Oceana in 2013) (*Warner et al., 2013*).

As always, these results rely on the quality of the reference database used, especially the identification of voucher specimens for reference sequences. The potential mislabeling we identified here requires further investigation in order to pinpoint the source of the substitution. Our study highlights the utility of using multiple analytical methods to identify specimens with standard DNA barcoding and especially that of statistical phylogenetics.

## ACKNOWLEDGEMENTS

We thank the Smithsonian's Laboratories for Analytical Biology (LAB) for lab space and sequencing services and the George Washington University's Colonial One High-Performance Computing Cluster for computational support.

### Funding

This work was supported by Whitfield Bryson & Mason LLP, Washington, D.C. Jason Rathod, then of Whitfield Bryson & Mason LLP, participated in discussions of the study design and results discussion and interpretation.

## Competing Interests

Keith Crandall is an Academic Editor for PeerJ. Jason Rathod is an employee of Migliaccio & Rathod LLP.

## Author Contributions

- David B. Stern performed the experiments, analyzed the data, wrote the paper, prepared figures and/or tables, reviewed drafts of the paper.
- Eduardo Castro Nallar analyzed the data, wrote the paper, reviewed drafts of the paper.
- Jason Rathod conceived and designed the experiments, wrote the paper, reviewed drafts of the paper.
- Keith A. Crandall conceived and designed the experiments, contributed reagents/materials/analysis tools, wrote the paper, reviewed drafts of the paper.

## Animal Ethics

The following information was supplied relating to ethical approvals (i.e., approving body and any reference numbers):

We sampled vertebrates from restaurants as food items and therefore no IRB approval was needed for the study.

## Data Availability

The Cytochrome Oxidase I sequences collected as part of this study are accessible via GenBank accession numbers KY656473–KY656484. Additionally, the Supplemental Information contains our fasta formatted data files for both the vertebrate and invertebrate data sets that include the aligned sequences from our study and other reference sequences from GenBank.

## Supplemental Information

Supplemental information for this article can be found online at http://dx.doi.org/10.7717/peerj.3234#supplemental-information.

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
