# Peer review of "DNA Barcoding analysis of seafood accuracy in Washington, D.C. restaurants"

_PeerJ, doi:10.7717/peerj.3234_

## Round 0.1 · original submission · Major Revisions

I have heard back from two reviewers, both of whom commented your paper is well written. One reviewer mentions the need for more information on the design and the phylogenetic analyses. In my opinion, these comments are valid but not too hard to address, so although my decision is "major revisions" I imagine this can be done with relative ease.

Reviewer 1 ·

Basic reporting

The manuscript if very well written, it is well explained.


Data will be shared before publication (submitted to GenBank already?).

Hypothesis, results and discussion agree.

Experimental design

The experimental design is correct.

Validity of the findings

Data is robust. The approaches described are very sound, and the analyses are the adequate for the task.

Conclusions are limited to the results, and discussion highlights the relevant points and consequences.

Additional comments

Some minor comments/typos:

1) Line 110: (Vrijenhoek 1994) is actually Folmer et al. 1994
Folmer, O., M. Black, W. Hoeh, R. Lutz, and R. Vrijenhoek. 1994. DNA primers for amplification of mitochondrial cytochrome c oxidase subunit I from diverse metazoan invertebrates. Molecular Marine Biology and Biotechnology 3:294-299.
Correct both in the text and in the references.

2) Line 112: the concentration of the dNTPs is missing: 0.5 μl dNTP (also, it is usually written as a plural, dNTPs.

3) Lines 199 and 241: Litopeneaus should read Litopenaeus

4) Line 244: “Using the character-based identification scheme of (Lowenstein et al. 2009), “ should read “Using the character-based identification scheme of Lowenstein et al. (2009),”

5) More a comment than asking for a change: I don’t personally think that character-based approaches are correct per se, since they strongly depend in our reference database (much more than phylogenetic and distance methods): they use nucleotide positions as “unique” for a species/species group based in the sequences available for that species/group. It is unknown if, within the species or group, that is variable or not (just within our individuals; and COI is a highly variable gene). But, that is a general flaw (in my opinion) of the character based identification scheme, and not this manuscript (that it is not about methods). But, my concern is the bias on this methodology, and consequences of their “positive/negative” results. In this case, it is supported by the phylogeny approach, that is what I think is the best approach in this case.

6) Line 284: Write out Farfantepenaeus. After a period, abbreviation for the genera should not be used. (. F. should read . Farfantepenaeus). Also, correct to Farfantepenaeus in line 282.

Reviewer 2 ·

Basic reporting

Clear and unambiguous, professional English used throughout - Yes

Literature references, sufficient field background/context provided - Yes

Professional article structure, figs, tables. Raw data shared - Yes

Experimental design

Original primary research within Aims and Scope of the journal - Yes

Research question well defined, relevant & meaningful. It is stated how research fills an identified knowledge gap - Partially - see general comments below

Rigorous investigation performed to a high technical & ethical standard - partially - see general comments below

Methods described with sufficient detail & information to replicate - partially - see general comments below

Validity of the findings

Data is robust, statistically sound, & controlled - partially - see general comments below

Conclusion are well stated, linked to original research question & limited to supporting results - partially - see general comments below

Additional comments

The present study is very important showing an interesting use of DNA barcoding methodology to identify mislabeled foods as other studies have been demonstrated. The authors apply different methodological approaches to try more strength results in your analyses, highlighting the potential use of phylogenetic approaches.
In general the study sounds good, but I have same notes. The authors do not make clear how many samples they are analyzed from ach restaurant. Was just one from each dish or more? Was taken only one sample or are there more samples in different days or from different batches? I think that this information is essential to give us an idea about the robustness of the analyses. If only one sample was taken, additional samples need to be included to increasing the accuracy of the analyses.
Other important point is the use of the phylogenetic approach to give more strength to the identification of each sample. As the authors said in lines 216-219, the use of only one sequence marker in a very distantly taxon is not adequate. Thus, should be interesting more explanation about this choice front of the weakness about it use for this specific issue. Beside this, presently are several other identification approaches developed to barcoding dataset more robust and powerful such as ABDG and GYMC that use the Maximum-likelihood method. Likewise, the use of the character-based identification is more adequate to this purpose, mainly among that species with low genetic divergence among them. This approach was used by the authors effectively. Thus, I strong suggest that the authors apply these analyses in their data, mainly ABDG and GYMC approaches which are more adequate to available dataset and their purposes.
Finally, I suggest that authors try to get some samples of the species with no available COI sequences in BOLD and NCBI, to produce the COI sequence reference to test its samples. This will improve the study.
Below, I point minor issues:
In the abstract (line 33) are pointed that six restaurants were visit, but in the material and methods sections are pointed that 12 restaurants were visited (line 106). Table 1 indicate six restaurants as well, please correct them.
Lines 125-126 – Actually, the BOLD systems have more COI sequences than NCBI and others sequences databanks. Many barcode projects in development deposit the COI sequences first in the BOLD Database which after send all of them to the NCBI Database.
Lines 126-127 – Please indicate what is the value of similarity were used in the top hits in both databases. These values are very important to understanding which similarity and divergences values you are working and considering.
Table 1 – indicate the BOLD ID values of target sequences.

---

## Round 0.2 · accepted · Accept

The revision has been well done, and this manuscript is now ready for publication. I look forward to seeing the published version of this work!

Reviewer 1 ·

Basic reporting

All questions have been properly addressed.

Experimental design

All questions have been properly addressed.

Validity of the findings

All questions have been properly addressed.

Additional comments

All questions have been properly addressed.